# Trait and plasticity evolution under competition and mutualism in evolving pairwise yeast communities

**ShengPei Wang[1], Renuka Agarwal[1], Kari A. Segraves[2], David M. Althoff[1]** *

1 Department of Biology, Syracuse University, Syracuse, New York, United States of America, 2 National Science Foundation, Alexandria, Virginia, United States of America

* dmalthof@syr.edu

**Data Availability Statement:** The data underlying the results presented in the study are available from Dryad.org. https://doi.org/10.5061/dryad.5qfttdzgf.

## Abstract

Although we have a good understanding of how phenotypic plasticity evolves in response to abiotic environments, we know comparatively less about responses to biotic interactions. We experimentally tested how competition and mutualism affected trait and plasticity evolution of pairwise communities of genetically modified brewer's yeast. We quantified evolutionary changes in growth rate, resource use efficiency (RUE), and their plasticity in strains evolving alone, with a competitor, and with a mutualist. Compared to their ancestors, strains evolving alone had lower RUE and RUE plasticity. There was also an evolutionary tradeoff between changes in growth rate and RUE in strains evolving alone, suggesting selection for increased growth rate at the cost of efficiency. Strains evolving with a competitive partner had higher growth rates, slightly lower RUE, and a stronger tradeoff between growth rate and efficiency. In contrast, mutualism had opposite effects on trait evolution. Strains evolving with a mutualist had slightly lower growth rates, higher RUE, and a weak evolutionary tradeoff between growth rate and RUE. Despite their different effects on trait evolution, competition and mutualism had little effect on plasticity evolution for either trait, suggesting that abiotic factors could be more important than biotic factors in generating selection for plasticity.

## Introduction

Species interactions are important drivers of evolution, exemplified by the phenotypic diversification of Darwin's finches [1], widespread aposematic coloration in butterflies [2], and active pollination behaviors in yucca moths and fig wasps [3, 4]. These remarkable examples of adaptation are caused by both direct selection from the interactions and indirect selection due to altered resource dynamics [5–8]. Given the natural variability in resource availability and interacting partners across time and space, many species have evolved plastic phenotypic expression in response to both abiotic and biotic factors [e.g., 9–11]. We define phenotypic plasticity as the ability of an individual to exhibit different phenotypes in response to changes in the local environment [12]. Although the importance of biotic interactions on trait

**Funding:** Grants from the U.S. National Science Foundation (DEB 1655544 and 2137554). The funders had no role in study design, data collection and analysis, decision to publish, or the preparation of the manuscript.

**Competing interests:** The authors have declared that no competing interests exist.

evolution has been demonstrated repeatedly [13–15], we are just beginning to understand how these interactions influence the evolution of phenotypic plasticity [12, 16].

Adaptive phenotypic plasticity has been shown to be important for many types of biotic interactions. For example, spadefoot toad larvae can develop either omnivore or carnivore morphs depending on prey density [17], and *Daphnia* individuals can plastically produce helmets that protect them from predators [18]. We also know that temporal and spatial heterogeneity in interaction types and strength can select for evolutionary increases in phenotypic plasticity [19], but little is understood about how biotic interactions differ in their effects on plasticity evolution. A notable exception is a simulation study by Scheiner et al. [16] that showed biotic interactions in general led to reduced plasticity when species are simultaneously adapting to abiotic and biotic conditions. Their results also showed that different types of biotic interactions varied in their effects on plasticity evolution. For example, antagonistic interactions such as predator-prey interaction had greater effects on the evolution of plasticity than competition or mutualism, while competition reduced plasticity in comparison to mutualism. Competition and mutualism both influence resource dynamics, but these two types of interactions change resource availability in opposing ways, making competition and mutualism strong starting points for investigating how biotic interactions centered on resources influence plasticity evolution. Phenotypic plasticity can affect not only the developmental programs of morphological traits, but also suites of physiological traits that influence resource use and life history strategies [e.g., 20, 21]. The latter is particularly important for organisms that live in variable resource environments such as seasonal changes in light and temperature, pulses of food, and temporal and spatial variability in niche parameters across a habitat [e.g., 22, 23]. For plasticity evolution, the key factor is whether resources are variable across space or time. All else being equal, competition would keep resources at consistently low levels. Mutualism may keep resources at consistently higher levels due to partner feedbacks, but individuals may still compete for access to partners or traded resources as populations of both mutualists grow. The effects of these two interactions will also be mitigated against the environmental input of resources such as through temporal and spatial pulses of resources. When limiting resources in the environment occur in distinct pulses, competitive interactions will quickly deplete those resources and lead to resource cycling from high to low. Mutualistic interactions could reduce temporal resource heterogeneity by providing a continual input of resources between pulses Thus, disentangling the importance of biotic interactions such as competition and mutualism from abiotic factors is necessary for understanding how phenotypic plasticity may evolve in interacting species [16].

The lack of empirical evidence partially stems from the challenge of monitoring plasticity evolution in species engaged in different types of interactions. To directly test the effects of biotic interactions, we need to make comparisons of similar species that consistently engage in different types of interactions—an extremely difficult experiment to conduct in natural systems. Microbial communities are good model systems that satisfy this need. Microbes can engage in diverse types of interactions, and their rapid generation time allows for monitoring of evolutionary changes in short time periods [24]. Although competition is the predominant interaction in some microbial communities [25], other types of interactions such as mutualism are also widespread [26, 27]. For example, syntrophic metabolism or microbial cross-feeding has been observed across many different habitats ranging from marine sediments to freshwater, and from thermal springs to permafrost [28]. Additionally, phenotypic plasticity has been shown to evolve rapidly in microbes in response to abiotic environments, both in terms of morphological and physiological traits [22, 29, 30]. Thus, experimental evolution with microbes is an accessible system in which to test the effects of species interactions on trait and plasticity evolution.

To understand the role of biotic interactions in the evolution of plasticity, we used yeast (*Saccharomyces cerevisiae*) strains from a synthetic experimental mutualist community [31, 32] and evolved them with either a competitive or a mutualistic partner strain. We quantified the evolution of two composite phenotypes, growth rate and resource use efficiency (RUE), as well as their plasticity. These two traits are universal across all organisms and can vary at within population, among population, and species levels as organisms adapt to the resource environment [33–35]. Growth rate should be selected to be maximized in all environments as it is also a strong proxy for fitness, especially in asexual organisms. Intricately tied to growth rate is the way in which organisms use resources (RUE). In high resource environments, organisms can more easily acquire the resources necessary for growth and RUE is not under as strong selection. In contrast, when resources are low and hard to acquire, increased RUE can lead to greater population growth. In general, faster growing and maturing individuals will outcompete slower growing individuals, but resource level influences what is the 'faster growing genotype'. Less efficient individuals may not obtain enough resources in low resource environments. For microbes, there is a tradeoff between growth rate and RUE in which fast growth comes at the cost of RUE [34, 36, 37], although this tradeoff may be resource level dependent [e.g., 38–40]. If growth rate and RUE change across different resource levels, it would suggest that microbes are responding to resource levels, either plastically or through evolution.

We also evolved all yeast strains alone to establish a baseline of evolution caused by the culturing conditions. In the absence of interspecific interactions, we expected the evolution of rapid growth, because growth rate is a direct measure of fitness in our serially propagated cultures [41, 42]. Fast growing cells are more likely to be represented in subsequent transfers, but these cells are usually less efficient and typically have a lower yield for a given amount of resources than slower growing cells [34, 37, 41]. Additionally, cycling in the availability of resources from abundant at the beginning of a transfer to extremely limited due to population growth and intraspecific competition should select for increased plasticity in resource use in order to maximize growth rate across all resource availabilities. Thus, we expected that growth rate per se would have reduced or no change across resource availabilities in evolved strains. Resource use efficiency as measured by yield should evolve to be lower because of the potential tradeoff with selection for fast growth.

Evolving with a competitor or mutualist may modify the resource dynamics and could change how traits and their plasticity evolve. Because competition causes resource limitation, we expected strains evolving with a competitor to respond similarly to strains evolving alone but to have a more pronounced evolutionary response—selection for even faster growth and reduced RUE. Plasticity for these measures will likely not evolve as the same pattern of variability in resources would be present as strains that are evolving alone and just being serially propagated. In contrast, mutualism can provide a continual input of limiting resources in which growth of one partner fuels the growth of the other partner and vice versa. Thus, the reduction of resources between pulses (transfers) would be smaller. Initially, population growth of each partner species will be mitigated by the availability of the traded resource. As mutualist populations grow, competition for other freely available resources in the environment will begin. If those resources are needed by both partner species, then they will likely compete for them as population sizes increase. With mutualism, however, there is a direct feedback to the partner species that is not present in solely competing species. If one partner is better at competing for additional shared resources and begins to exclude its mutualistic partner, this will reduce the availability of the traded resource and impact the better competitor. Because of this, there is an inherent push and pull of providing resources for your partner species, but not completely outcompeting them. Such a scenario is potentially important in many

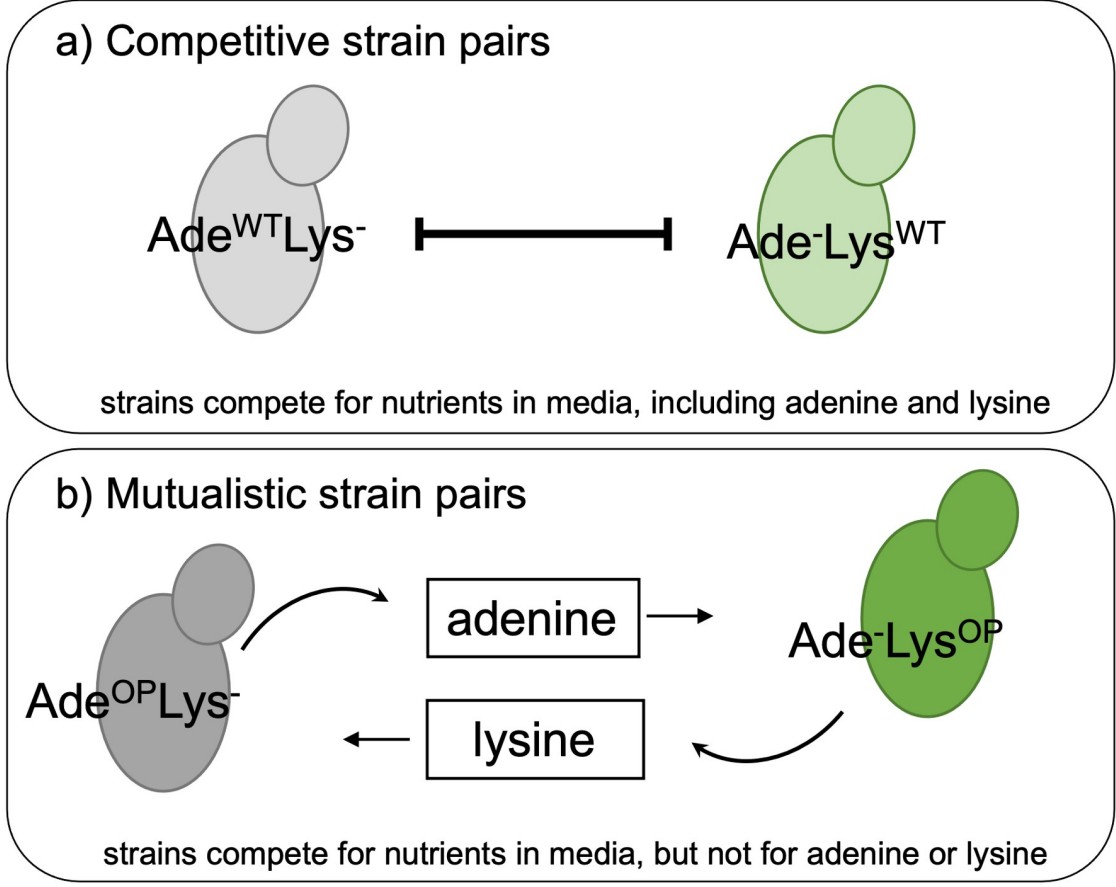

**Fig 1. Schematic representing community combinations of *S. cerevisiae* strains that differ in production of adenine and lysine.** Grey colors denote strains that are produce adenine at normal, wild type levels or overproduce adenine but are deficient in producing lysine, and green colors denote strains that produce lysine at normal, wild type levels or overproduce lysine but are deficient in producing adenine. Strains were paired with a competitor (a) or mutualist (b) to examine the effects of biotic interactions on evolution of growth rate, resource use efficiency, and their plasticity. The single arrows represent uptake from the environmental pool of adenine or lysine, and curved arrows represent overproduction of either adenine or lysine that is released into the environmental pool.

microbe communities that have very similar general resource requirements such as gut microbes. What this means for the evolution of growth rate and RUE is unclear.

To construct competitive and mutualistic communities, we combined yeast strains with different mutations in key metabolic genes (Fig 1). Specifically, the competitive strains were deficient in production of one essential nutrient, either adenine (Ade) or lysine (Lys) but were self-sufficient to produce the other nutrient (Ade$^{WT}$Lys$^-$ and Ade$^-$Lys$^{WT}$), whereas the mutualists strains were similarly deficient in producing one of the nutrients but overproduced an excess of the other nutrient (Ade$^{OP}$Lys$^-$ and Ade$^-$Lys$^{OP}$). We chose media concentrations such that a competitive pair needed to compete for resources other than adenine and lysine, and that a mutualist pair benefits from each other's adenine and lysine production while also competing for other resources. After 4 weeks of evolution, we compared the ancestral and evolved strains in their growth rate, RUE, as well as their phenotypic plasticity in response to resource availability. Adenine and lysine have different functions in cells. Adenine is important in cellular respiration and nucleotide synthesis and lack of adenine prevents yeast cell division [42].

Lysine is important in protein synthesis and responding to oxidative stress, and lack of lysine results in increased susceptibility to oxidative stress [43] and apoptosis [44]. Vidal et al. [31] demonstrated that lysine deficient strains ($Ade^{OP}Lys^-$ and $Ade^{WT}Lys^-$) die at a faster rate than adenine deficient strains when starved for the corresponding essential nutrient. This asymmetry in nutrient function may influence evolutionary changes in resource use, growth and population dynamics for different strain types.

Using these strains and experimental design, we addressed three specific questions: (1) How strains evolving alone respond to the environmental conditions during culturing (i.e., the abiotic environment)? (2) How does evolving with a competitor change the pattern of evolution as compared to evolving alone? (3) How does evolving with a mutualist change the pattern of evolution as compared to evolving alone or with a competitor? Our goal was to determine if evolving with different types of interspecific interactions changes the overall response of trait and plasticity evolution above and beyond responding solely to the abiotic conditions.

## Methods

### Community setup

We used eight genetically engineered and reproductively isolated strains of *Saccharomyces cerevisiae* [31] to form two types of communities dominated by either mutualistic or competitive interactions (Fig 1, S1 Table). The differences in the strains' ability to produce essential nutrients determine how they interact. The competitor strains are self-sufficient with either adenine or lysine but are deficient in producing the other nutrient (CA: $Ade^{WT}Lys^-$ and CL: $Ade^-Lys^{WT}$), thus, in medium with lysine and adenine available, they interact competitively for all essential nutrients. In contrast, the mutualist strains provide complementary benefits to one another, and the partners overproduce either adenine or lysine and are deficient in making the other nutrient (lysine and adenine, respectively) When the two types of mutualist strains (MA: $Ade^{OP}Lys^-$ and ML: $Ade^-Lys^{OP}$) are grown together, they benefit from each other's overproduction [31]. In these communities, the strains are initially limited by access to the resource produced by the partner strain. Once population sizes increase, other nutrients will become limiting and competition will occur for those nutrients. We used additional genetic markers—histidine and leucine deficiency—to differentiate interacting strains within co-cultures using positive selection on agar plates. We created strain pairs of all possible combinations of interaction types and marker types to account for potential marker effects. In total, we used four pairs of interacting strains to test the differences between mutualistic and competitive communities: MA his- with ML leu-, MA leu- with ML his-, CA his- with CL leu-, and CA leu- with CL his-. Since these strains are reproductively isolated and can perform different ecological roles, we treat them as different species. They also differ in growth rate and yield at 24 hrs. (S1 Fig). Strains that produce adenine at wildtype levels or overproduce adenine have higher growth rates than strains deficient in adenine production. In contrast, strains that produce lysine at wildtype levels or overproduce lysine have higher yield at 24 hrs.

### Experimental design

We used experimental evolution to investigate the effect of biotic interactions on the evolution of phenotypic plasticity. We set up three treatment levels of biotic interactions: communities initiated with a single strain (evolved alone with no interspecific interactions), competitor pairs, and mutualist pairs, giving a total of 12 unique community types (S2 Table). The communities with strains evolved alone allowed us to test for the selective effects of the culturing environment. The culturing medium was modified from a standard yeast medium (0.15% (w/v) Difco yeast nitrogen base without amino acids or ammonium sulfate, 0.5% ammonium

sulfate, 2% (w/v) dextrose, with supplemental amino acids), and we reduced the adenine and lysine content by 40% (to 24mg/L of adenine; 54mg/L of lysine). This reduced level of adenine and lysine allowed all experimental communities to sustain growth while the mutualistic communities reached higher total biomass than competitive communities.

Experimental communities were maintained in 2 ml of medium in 48-well deepwell plates. The plates were covered with aluminum foil punctured once with a sterile needle for aeration and placed on a rotating wheel at 30°C under 24h darkness. We started with 75 replicates of each community type and obtained data on a total of 166 communities (75 single, 56 mutualistic, and 35 competitive) because some communities went extinct during the experiment. Extinction of paired strain communities was likely due to a combination of slight differences in growth rates among Ade⁻ and Lys⁻ strains, the serial dilution process itself, and interaction type. For example, the serial dilution process produces a population bottleneck during the transfer event. Any factor such as competition, reduced population size of mutualist partner or sampling error that reduces the size of the transferred populations will start a downward trend in a strain's population size. This process gets compounded during the serial dilution process as only 10% of the total community is transferred. The fact that single strain communities never went extinct suggests that interacting with a partner strain places additional selective pressure on strains that will influence long term persistence. The use of similar null alleles for the competitor strains and mutualist strains served as a control for initial strain differences. All cultures were initiated at a low population density of 0.1 $OD_{600}$ with equal starting densities for strain pairs. The clones used to initiate the experiment were flash frozen in 25% glycerol and stored at -80°C. These were designated as the 'ancestral' strains. Cultures grew for two days before the first transfer and were then transferred to fresh medium daily. We used a standard volume transfer of 5% (100 μL). We allowed the experimental cultures to evolve for four weeks, and at the end of the four weeks, the cultures were frozen as above. These cultures were designated as 'evolved'.

### Growth measurements

We assayed growth features of the ancestral and evolved strains by assaying monocultures in two media types. We first revived the ancestral and evolved strains by plating the frozen cultures onto selective agar plates where only one type of strain can grow (four types of selective plates with standard medium: -Ade-Leu for $Ade^{WT/OP}His^-$, -Ade-His for $Ade^{WT/OP}Leu^-$, -Lys-Leu for $Lys^{WT/OP}His^-$, and -Lys-His for $Lys^{WT/OP}Leu^-$). To ensure that all replicates had independent evolutionary histories for the evolved strains, we haphazardly picked one colony per strain type per community and grew it for two days in the same liquid medium that was used during experimental evolution (with daily transfer from low density, determined visually). The two days of growth helped to standardize subsequent trait measurements, because it allowed all the inoculating cultures to reach similar and relatively high population sizes prior to setting up the assay cultures for growth measurements. We used the day two cultures to inoculate two assay cultures in two nutrient environments: low versus high adenine and lysine (4mg/L of adenine and 9mg/L of lysine for the low environment, and, 24mg/L of adenine and 54mg/L of lysine for the high environment) starting from a low density (0.1 $OD_{600}$). This setup allowed us to assess both trait evolution in the original evolutionary environment (high) and a second environment in which to determine trait plasticity. We grouped strains by their initial strain type, (MA: $Ade^{OP}Lys^-$, ML: $Ade^-Lys^{OP}$, CA: $Ade^{WT}Lys^-$, or CL: $Ade^-Lys^{WT}$) for these growth assays so that measurements of ancestral and evolved strains from each treatment were done together and directly comparable (S2 Table). Each group of strains contained one evolved strain from all possible treatments and two colonies of the ancestral strain. For example, an

ancestral MA strain and all of its evolved MA descendants were measured at the same time. All cultures were initiated as 2ml cultures in 48-well deepwell plates as described above and were aerated on a rotating wheel in a dark room at 30 ˚C.

We measured growth rate of the assay cultures during the exponential phase of population growth and yield at 24 hrs. To measure growth rate, we measured population density (as determined by $OD_{600}$) at 4, 6, and 24 hours. We calculated growth rate, *r*, as the number of doublings during exponential growth between 4–6 hours:

$$r = \ln\left(OD_{600}^{t1}/OD_{600}^{t0}\right)/\ln(2)$$

We used the $OD_{600}$ value at 24 hours as the measure of yield. We chose this time frame because this was the duration between transfers for the evolution experiment and represented the greatest population size attainable before transfer. Previous work [31] also demonstrated that there are very few dead or dying cells at this time point so that optical density measurements would accurately depict the size of the growing population.

## Resource use efficiency measures

We quantified resource use efficiency (RUE) by using yield as a proxy. Our yield measures can be interpreted as the biomass produced given the amount of resources in the media. In both the low and high assay environments, yield was limited by either adenine and lysine and represented adenine or lysine use efficiency. Thus, within an assay environment, more efficient strains achieve higher population yield. The yield measures in the two assay environments were not directly comparable as efficiency measures, because the high environment contained six times more adenine and lysine than the low environment. Thus, we divided the yield measures from the high assay environment by 6 to standardize them to the low nutrient assay environment, making all RUE measures represent the amount of biomass produced with 4mg/L of adenine and 9mg/L of lysine.

Because there is usually a tradeoff between growth rate and RUE in microbes, we also tested for this possibility during the course of the evolution experiments. For the strains evolving alone, we calculated the change in growth and RUE between the ancestor and evolved strained. For the strains evolving with a competitor or mutualist, we calculated the change in growth and RUE between the strain evolved alone and the strain evolved with a partner. We then examined the relationship between the change in growth rate and the change in RUE.

## Plasticity measures

We calculated plasticity of growth rate and RUE for each clone as the difference in the trait value between the low and the high adenine and lysine assay environments. We calculated the growth plasticity as growth rate measured in the high minus the low environment (growth $rate_{high}$−growth $rate_{low}$). For efficiency plasticity, however, we used efficiency measured in the low minus the high environment for efficiency plasticity ($efficiency_{low}$−$efficiency_{high}$) because efficiency decreased with increasing environmental adenine and lysine.

## Data analysis

We used the differences in trait and plasticity values between strains evolved alone and their ancestors to quantify the effects of the abiotic environment (evolving under our culturing condition for 4 weeks), and we used the differences between strains evolved alone and strains evolved with either a competitive or mutualist partner to quantify the effect of biotic interactions. For example, we calculated plasticity for an ancestral strain and a strain evolved alone,

and then we subtracted the ancestral value from the evolved value to determine the change in plasticity. A negative value would indicate that plasticity evolved to be lower, i.e., the evolved strain was less responsive to nutrient concentrations. These trait and plasticity differences were calculated between pairs of strains that were measured at the same time, which allowed us to remove some of the random variation caused by genotypes and measurement batches. Since each measurement batch contained only one evolved strain from each genotype and treatment, calculating the trait differences (evolved alone minus ancestors, and evolved with a partner minus evolved alone) did not compromise independence or replication but yet allowed us to simplify our models.

We used linear mixed models to analyze the effects of the abiotic culturing conditions (evolved alone minus ancestors) and the effects of biotic interactions (evolved with a partner minus evolved alone) for growth rate, growth plasticity, RUE, and RUE plasticity. We analyzed changes in growth rate and RUE using measurements from only the high adenine and lysine environment, which was the same as the evolutionary environment. We first tested whether the genetic differences among the experimental strains were important by using models containing the strain type (either Ade⁻ or Lys⁻) and the marker genotype (His⁻ or Leu⁻) and their interaction as covariates. These genetic effects were minimal for all models, so we simplified all the models to have only strain identity as a random effect to simplify model interpretation. Given this result, we then pooled data on trait values from strains used in each of the evolutionary treatments. In the comparison of trait values between ancestral strains and those evolved alone, we calculated average traits values using all strain types, (MAs, MLs, CAs, CLs). For comparisons between strains evolved alone or with a competitor we pooled data from the CA and CL strains, and between strains evolved alone or with a mutualist we pooled data from the MA and ML strains.

We fitted four models (one for each trait) to test whether the evolutionary effect was different from zero, represented by the intercept in the model output (abiotic effects in Table 1). Similarly, we fitted another four models to test the effect of biotic interactions with the type of interaction as a fixed effect; thus, the intercept represented the effects of competition in comparison with strains evolved alone, and the slope represented the effects of mutualism in comparison with competition (marked as competition and mutualism in Table 1). All models had strain identity as a random effect.

**Table 1. Results of experimental evolution of *S. cerevisiae* strains growing in culture with and without biotic interactions.** Model outputs for models testing effects of evolution in response to the abiotic culturing condition (shaded) and for models testing effects of evolution with a competitive or mutualistic partner (no shading). Tests compare whether the type of interaction changes the pattern of evolution as compared to the abiotic treatment.

| | Effect type | Estimate | SE | p-value |
|---|---|---|---|---|
| **Growth rate** | Abiotic | 0.004 | 0.004 | 0.35 |
| | Competition | 0.015 | 0.008 | **0.05** |
| | Mutualism | -0.017 | 0.010 | 0.09 |
| **Growth plasticity** | Abiotic | 0.005 | 0.006 | 0.40 |
| | Competition | 0.016 | 0.009 | 0.08 |
| | Mutualism | -0.014 | 0.012 | 0.24 |
| **RUE** | Abiotic | -0.117 | 0.038 | **0.02** |
| | Competition | -0.116 | 0.060 | 0.06 |
| | Mutualism | 0.193 | 0.076 | **0.01** |
| **RUE plasticity** | Abiotic | -0.057 | 0.022 | **0.04** |
| | Competition | -0.018 | 0.041 | 0.68 |
| | Mutualism | 0.032 | 0.052 | 0.56 |

We also investigated the evolutionary tradeoff between growth rate and efficiency using simple linear regression. We used the trait differences in growth rate between pairs of strains described above as the predictor variable and the trait differences in RUE as the response variable. A negative correlation between these two measures suggested an evolutionary tradeoff between growth rate and RUE.

For each trait, we compared the same strain evolving alone to that strain evolving with an interacting partner in the same abiotic environment. This controlled for the effects of evolution to the culturing conditions [45]. All statistical analyses were done in the R environment [45]. We used the lme4 and lmerTest packages for fitting the linear mixed models and the ggplot package for figures [46, 47]. For interpreting the model outputs, we used a significance threshold of 0.05.

## Results

### Ancestral patterns in growth rate, RUE and plasticity

To provide a baseline for trait evolution, ancestral strains were grown individually in both media types and measured for population size at 4, 6 and 24 hrs. via optical density. Growth rate was determined by the difference in growth between 4 and 6 hrs. All ancestral strains exhibited differences in growth rates and RUE across the two assay environments. Strains grew more slowly when placed in the low resource environment as compared to the high resource environment (Fig 2a), and strains had higher RUE in the low resource environment as compared to the high resource environment (Fig 2b). These changes across resource environments meant that ancestral yeast strains responded to the resource environments and thus are plastic with regard to resource input. With this baseline, the next step was to understand whether the evolutionary treatments changed trait values and the ability of yeast strains to respond to changing resource environments.

### How does evolution in isolation of interspecific biotic partners influence the evolution of traits related to fitness?

Strains were grown alone to understand how the culturing environment affected trait evolution, and these evolved strains were used as the control with which to compare the effects of competition and mutualism (Fig 2). Strains evolved alone had similar growth rates to their ancestors (Table 1). Similarly, there was no statistically significant change in growth rate plasticity across the low and high adenine and lysine media environments (Table 1). In contrast, strains evolved alone had statistically lower RUE (-7.9%) and RUE plasticity (-30.1%) than their ancestors (Table 1). These comparisons provided the baseline expectations of evolutionary changes due to the culturing conditions used in our experiment—no change in growth rate or growth rate plasticity, but lower RUE and a decrease in RUE plasticity.

### How does evolving with a competitor change the pattern of evolution in growth, efficiency, and their plasticity?

In comparison to strains evolved alone, surviving strains evolved with a competitor had significantly higher growth rates, marginally significant lower RUE, and no difference in either plasticity measures (Table 1; Figs 2–4). Resource use efficiency was 3.6% lower for strains evolved with a competitive partner (Table 1; Figs 2, 3c and 4b). There was little difference in growth plasticity or RUE plasticity between strains evolved with a competitive partner or those evolved alone (Table 1; Figs 2, 3b, 3d and 4d). One of the CA strains (1066) went extinct late in the experiment in all the cultures when it was paired with a competitor strain.

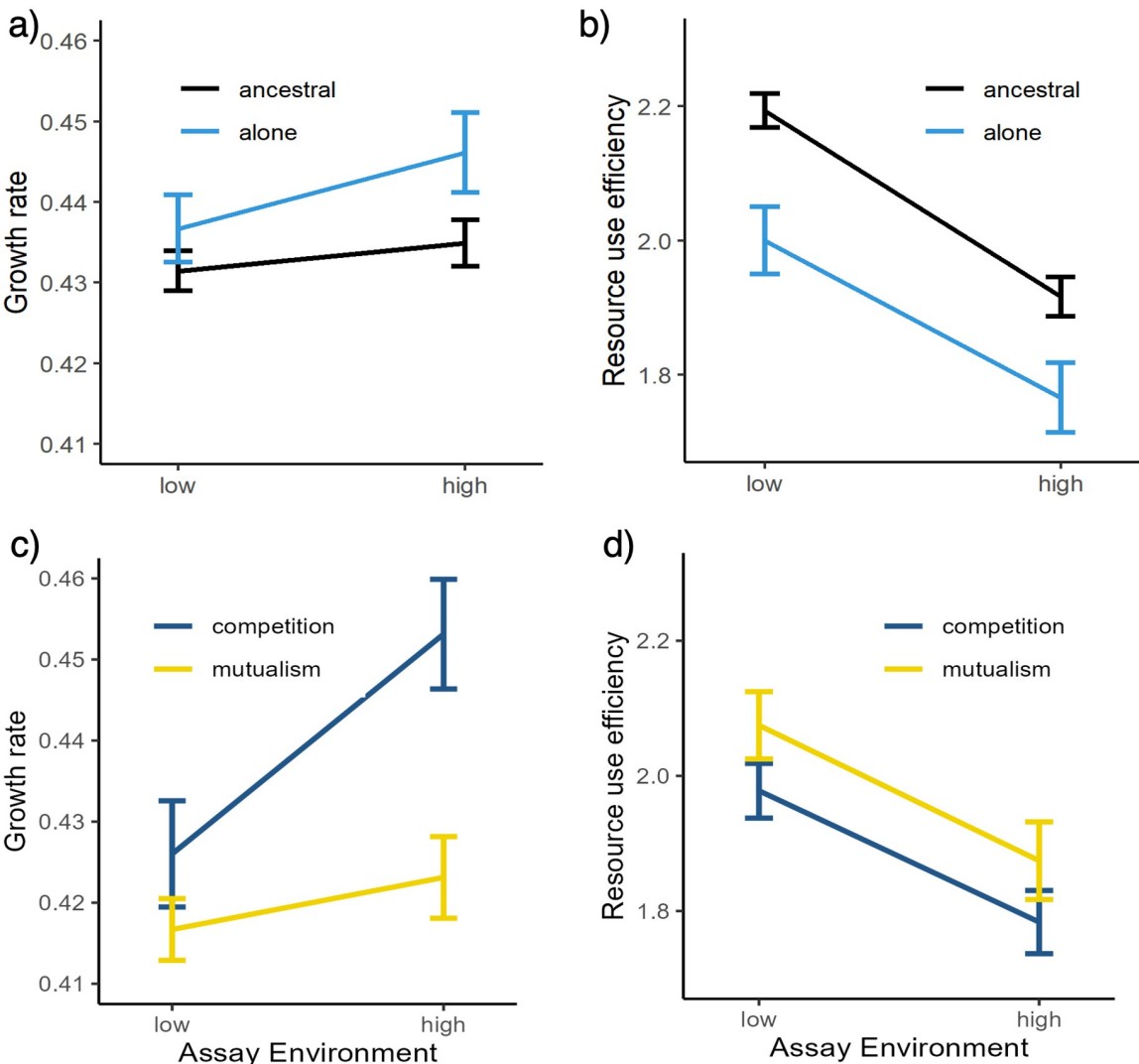

**Fig 2. Reaction norms of growth rate (a, c) and resource use efficiency (b, d) for ancestral strains, strains evolved alone, strains evolved with a competitive partner, and strains evolved with a mutualistic partner.** Data from all strains in each treatment and community context were pooled together as there were no differences among strains within each treatment. Statistical comparisons of trait evolution between evolving with a competitor or a mutualist versus alone are reported Table 1.

## How does evolving with a mutualist change the pattern of evolution in growth, efficiency, and their plasticity?

Strains evolved with a mutualist partner had similar growth rates and plasticity in growth rate as strains evolved alone, although there was trend for slower growth (Table 1, Figs 2, 3a, 3b, 4a and 4c). Evolving with a mutualistic partner, however, selected for increased RUE but not RUE plasticity (Fig 4b and 4d). In contrast to strains evolved with a competitor, strains evolved with a mutualistic partner had 5.6% higher resource use efficiency (Table 1, Figs 2, 3c and 4b). These results demonstrate that interaction type may lead to different selective pressures in terms of changes in the evolution of RUE.

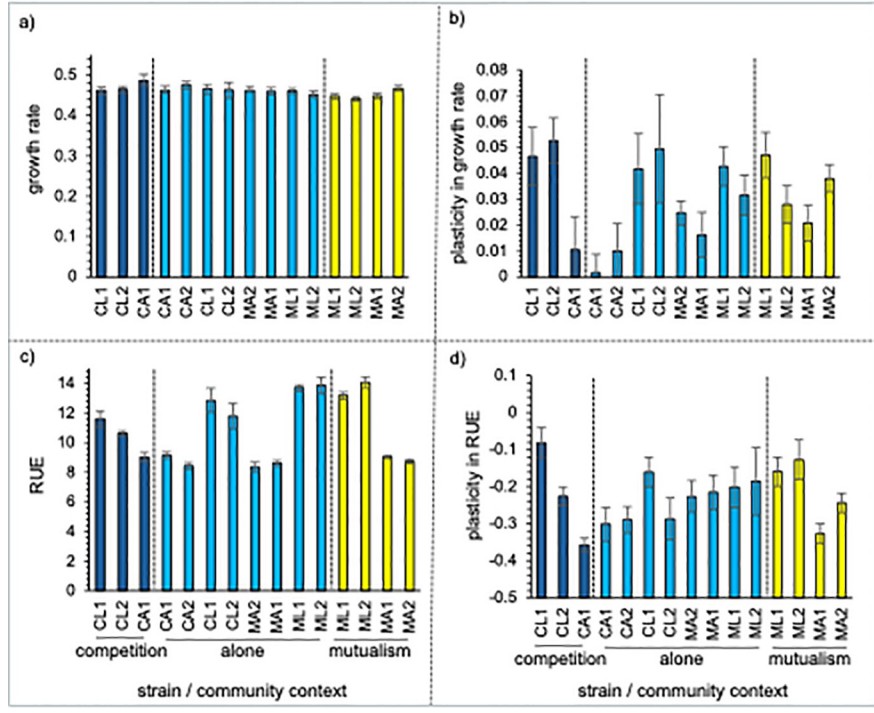

**Fig 3. a) Growth rate, b) growth rate plasticity, c) resource use efficiency (RUE), and d) RUE plasticity for yeast strains grown alone (light blue), under competition (dark blue), or under mutualism (yellow).** Plasticity was measured as the difference in trait values from growing alone in a low or high resource environment. CA2 strains went extinct when in competition with CA1 strains. Error bars are standard errors. Overall tests for effects of community context on trait and plasticity evolution in Table 1.

## Tradeoffs in growth and efficiency

We found strong evolutionary tradeoffs between growth rate and RUE among strains evolved alone and strains evolved with a competitive partner, but not in strains evolved with a mutualistic partner. Comparing strains evolved alone to their ancestors revealed that there was a strong negative correlation between evolutionary changes in growth rate and changes in RUE ($\beta$ = -3.4 ± 0.9, p < 0.01, Fig 5a). A similar pattern was observed in strains evolved with a competitor ($\beta$ = -4.6 ± 1.3, p < 0.01, Fig 5b). In contrast, this negative correlation was lower and not statistically different from zero for strains evolved with a mutualistic partner ($\beta$ = -1.5 ± 1.3, p = 0.27, Fig 5c).

## Discussion

The phenotype of an organism is shaped by the interaction between its genetics and the environment, i.e., its phenotypic plasticity. Phenotypes of interacting organisms determine the outcomes of their interactions with one another which then can influence their fitness. Furthermore, through biotic interactions, phenotypic plasticity can influence a population's ecological success and evolutionary trajectory [reviewed 12]. Some studies have demonstrated that plasticity can evolve quickly in response to variation in biotic partners [11, 48]; however, we know very little about whether different types of biotic interactions tend to impose similar or contrasting effects on phenotypic and plasticity evolution. Using a synthetic yeast model,

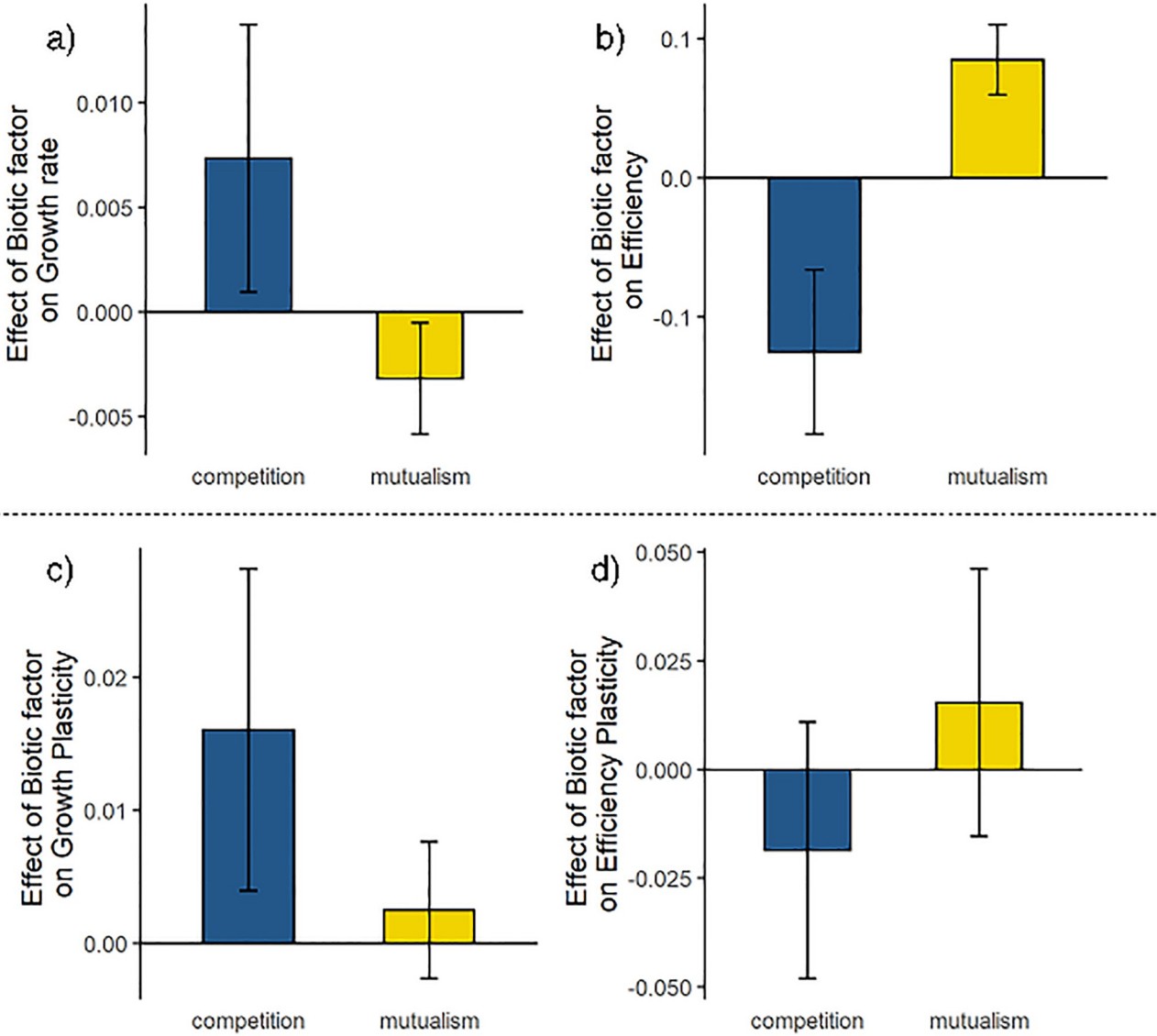

**Fig 4. Average effects of competition and mutualism on a) growth rate, b) resource use efficiency, c) growth plasticity, and d) resource use efficiency plasticity.** Error bars represent standard errors. Values close to zero means that the trait values of strains evolved with a biotic partner are similar to strains evolved alone. Asterisks indicate statistical difference from the strains evolved alone. Data for each biotic interaction were pooled across all strains involved (i.e, all mutualist strains, all competitor strains).

we found that different types of biotic interactions led to contrasting trait evolution but similar plasticity evolution. Specifically, competitive interactions led to the evolution of increased growth rate, reduced resource use efficiency (RUE), and a strong tradeoff between growth and RUE. In comparison, mutualism led to increased RUE and no tradeoff between growth and RUE. Despite the observed evolutionary changes in trait values, neither competition nor mutualism had much of an effect on the evolution of growth plasticity or RUE plasticity. Together, these results show that different types of species interactions can alter the environment in such a way to drive trait evolution in opposing directions, but overall, biotic interactions were less important for plasticity evolution.

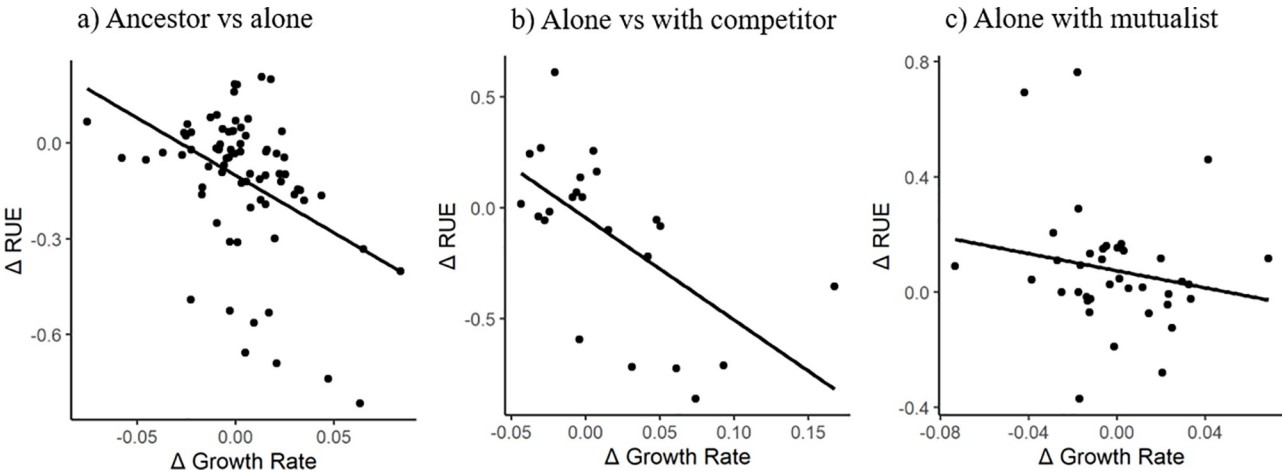

**Fig 5.** Relationships between Δ growth rate and Δ resource use efficiency between (a) ancestral strains vs. strains evolved alone (β = -3.4 ± 0.9, p < 0.01, r = -0.42, P < 0.0002). (b) A similar pattern was observed in strains evolved with a competitor (β = -4.6 ± 1.3, p < 0.01, r = -0.61, P < 0.0019). (c) In contrast, no relationship was detected for strains evolved with a mutualistic partner (β = -1.5 ± 1.3, p = 0.27, r = -0.18, P > 0.05). A negative relationship shows an evolutionary tradeoff between growth rate and efficiency.

### Baseline evolutionary patterns of adapting to the local abiotic environment

Although we expected growth rate and growth plasticity to evolve as a result of culturing conditions for our yeast communities, strains that evolved alone did not show a significant change in either trait. We expected the evolution of increased growth rate in our evolved strains because growth rate should contribute positively to fitness in well-mixed cultures that are frequently transferred. Our transferring regime involved daily nutrient inputs that should have selected for rapid growth during the period of resource abundance when most population growth happens [41]. One potential reason for the stasis in growth rate and its plasticity is that our serial propagation regime and the medium are very similar to the conditions under which the ancestral strains have been maintained in general. Thus, growth may have been already optimized and an additional 4 weeks of evolution was insufficient to change growth or its plasticity.

In contrast to growth rate, evolution in the abiotic culturing environment did cause a reduction in resource use efficiency. Reduction in efficiency is a common response of microbes to serial transfer culturing methods and is driven by selection to obtain resources and reproduce as quickly as possible to have propagules that survive the next transfer event and population bottleneck [41, 49]. The abiotic environment also had an effect on RUE plasticity evolution. Although evolved strains did exhibit plasticity in RUE (Fig 3b), plasticity evolved to be lower than ancestral strains, suggesting that strains evolved alone were less responsive to changes in resource concentration, likely as a result to maximize growth rate. In addition to changes in individual traits, we also detected a strong negative tradeoff between growth rate and RUE (Fig 4a), suggesting that the abiotic environment selected for fast growing, less efficient cells to maximize the probability of having propagules during serial transfer. Additionally, there was more variance in traits values from strains evolved alone when compared to ancestrals, indicative of the fact that each replicate was free to evolve along a different trajectory (Fig 2a and 2b). This result suggests that evolution in response to an interacting partner would have to be strong enough to overcome this additional noise added by replicate communities evolving independently from one another. Overall, the results from strains evolved alone

confirmed that our experimental design was sufficient to monitor evolutionary changes in both traits and trait plasticity.

## Evolutionary patterns when interacting with partners differing in interaction type

There were consistent evolutionary patterns between strains evolved alone and those evolved with a competitive partner, suggesting that intra- and inter-specific competition had similar effects on trait evolution in our experiment. We predicted that strains evolved with a competitor strain should have similar, but more pronounced changes in the measured traits as compared to strains that grew alone. Indeed, strains that evolved with a competitor had significantly higher growth rates than strains evolved alone (Fig 4a), and there was a slight increase in growth plasticity, albeit this effect was only marginally significant (p = 0.08, Table 1). The tradeoff between growth rate and efficiency was also more pronounced under competition, suggesting increased selection to forage for and use resources as quickly as possible (Fig 5b). Taken together, these results suggest that interspecific competition served to further increase the selection pressures that were present in the culturing conditions: to obtain resources and reproduce as quickly as possible.

Unlike competition, mutualism had distinctive effects on trait evolution. Strains evolved with a mutualist partner had slightly lower growth rates and significantly greater efficiency as compared to strains evolved alone (Table 1; Fig 4a and 4b). In terms of plasticity, however, these strains were not different from those that evolved alone. Notably, the effects of mutualism for all four traits were the opposite of the effects of competition (Table 1). The distinct patterns of trait evolution in mutualist versus competitive communities may have been driven by the different patterns of resource dynamics caused by these interactions. The continual input of a limiting resource from a mutualistic partner may have favored lower growth rates and increased efficiency. Specifically, in resource exchange mutualisms in which mutualistic resources are freely available in the local environment, temporal heterogeneity in resource availability may be less pronounced due to the continuous input of resources by the mutualistic partners. In the mutualistic communities used in this study, yeast strains continually added either adenine or lysine above and beyond the amount supplied in the medium. The addition of these resources by the mutualists would provide continual nutrient inputs that would delay the point at which adenine and lysine were exhausted. Under these conditions, the culturing conditions become more similar to chemostat conditions that select for slower growth rates and more efficient use of resources that lead to higher yields [41]. Additionally, the boost of nutrients from the mutualism could shift resource limitation from adenine and lysine to other nutrients, thus altering patterns of evolution in the mutualistic communities. Our results supported these expectations, demonstrating the contrasting effects between competition and mutualism.

The different selective environment created by mutualism also changed the evolutionary tradeoff between growth and efficiency. Specifically, there was an evolutionary tradeoff between growth and efficiency among strains evolved alone and with competitive partners, but this tradeoff was not observed for strains evolved with mutualistic partners. In the competitive communities, resources were added in distinct pulses that were rapidly depleted by population growth immediately after transfer to fresh media. This strongly heterogeneous resource environment likely selected for rapid conversion of resources into offspring and favored fast growth at the cost of reduced efficiency, leading to a strong evolutionary tradeoff between growth and efficiency. Similarly, the daily transfers also produced resource heterogeneity in the mutualistic communities, where abundant resource conditions after each resource pulse

favored rapid growth. After the initial pulse was depleted, however, the mutualism would have maintained a low abundance of adenine and lysine that supported continual population growth and increased resource use efficiency. The two alternating situations in the mutualist communities favored growth and efficiency at different times, which probably served to attenuate the evolutionary tradeoff between growth and efficiency.

Although the effects of competition and mutualism on plasticity were not significant, their effects were opposite to each other for both growth rate plasticity and RUE plasticity (Table 1). These findings are consistent with a theoretical study in which Scheiner et al. [16] found that although biotic interactions differed in their effects on plasticity evolution, biotic interactions in general led to reduced plasticity when interacting species are simultaneously adapting to abiotic conditions. Both their predictions and our results suggest that abiotic factors may exhibit stronger selection on plasticity than biotic factors such as competition and mutualism. However, the generality of this pattern is unclear as there are examples of the role of predation and herbivory driving induced defenses [50, 51]. Future research should directly test the relative importance of abiotic and biotic factors on plasticity evolution across a range of species interactions and environments.

Although our experiment showed that biotic interactions are important for trait evolution, our interpretations are limited to these simple communities involving only two species. Natural communities, on the other hand, will involve many more species, and can involve species with very different ecologies, including differences in resource requirements, life history, and generation times. In addition to these biotic factors, temporal and spatial variation in abiotic environments can also modulate outcomes of biotic interactions and plasticity evolution. Although not addressing these potentially important factors, our results highlight the utility of using microbial experimental evolution to test how biotic interactions influence trait evolution.

## Conclusions

We found that different types of biotic interactions influenced trait evolution in drastically different ways but had little effect on plasticity evolution. Exploitative competition, either intra- or inter-specific, led to a reduction in RUE. In contrast, resource-exchange mutualism selected for a different life-history strategy that included selection for higher RUE. This contrast between competition and mutualism was potentially due to differences in the temporal dynamics of resource availability in these communities. Unlike trait evolution, competition and mutualism had little effect on the evolution of either growth rate plasticity or RUE plasticity after 4 weeks of evolution. This finding suggests that abiotic factors may be more important than biotic factors in favoring plasticity; however, this is likely context dependent. More research is needed to address the relative importance of abiotic and biotic factors on plasticity evolution and to provide a general framework for the role of species interactions in generating plasticity.

## Supporting information

**S1 Fig. Ancestral values for growth and yield used in experimental evolution experiments.** Strains deficient in adenine production (CLs or MLs) had significantly reduced growth rates ($F_{3,188} = 17.84$, P <0.0001), but higher population sizes at 24 hrs ($F_{3,188} = 660.67$, P < 0.0001, P <0.0001). Error bars are standard errors. Grey colored bars are strains that produce lysine at wildtype (light grey) or overproduce lysine (dark grey) and green colored bars are strains that produce adenine at wildtype levels (light green) or overproduce adenine (dark green). (PDF)

**S1 Table. Strain identification and genotypes for strains used in experimental evolution experiments.**
(DOCX)

**S2 Table. Experimental communities and strain pairings used in experimental evolution experiments.** Strains with different genetic markers were used to facilitate separation on selection plates.
(DOCX)

## Acknowledgments

The authors thank G. Bode, A. Curé, T. Johnson, N. Mohan Babu, K. Thomas, M. Vidal for helpful comments and discussions to improve the manuscript. We also thank D. Kishore and an anonymous reviewer for providing detailed comments and excellent suggestions that improved the manuscript. This publication is based on work conducted while KAS served at the National Science Foundation.

## Author Contributions

**Conceptualization:** ShengPei Wang, Renuka Agarwal, David M. Althoff.

**Data curation:** ShengPei Wang.

**Formal analysis:** ShengPei Wang, David M. Althoff.

**Funding acquisition:** Kari A. Segraves.

**Methodology:** ShengPei Wang, Kari A. Segraves, David M. Althoff.

**Supervision:** David M. Althoff.

**Writing – original draft:** ShengPei Wang, David M. Althoff.

**Writing – review & editing:** ShengPei Wang, Renuka Agarwal, Kari A. Segraves.

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
