## [Decision Letter · Decision Letter 0]

6 Jun 2024

PONE-D-24-10370Trait and plasticity evolution under competition and mutualism in evolving pairwise yeast communitiesPLOS ONE

Dear Dr. Althoff,

Thank you for submitting your manuscript to PLOS ONE. After careful consideration, we feel that it has merit but does not fully meet PLOS ONE’s publication criteria as it currently stands. Therefore, we invite you to submit a revised version of the manuscript that addresses the points raised during the review process.

We look forward to receiving your revised manuscript.

Kind regards,

Karthik Raman, Ph.D.

Academic Editor

PLOS ONE

Journal Requirements:

Grants from the U.S. National Science Foundation (DEB 1655544 and 2137554).

The authors thank G. Bode, A. Curé, T. Johnson, N. Mohan Babu, K. Thomas, M. Vidal for helpful comments and discussions to improve the manuscript. Funding for this project was provided by a Sigma Xi Grant in Aid of Research to SW and grants from the U.S. National Science Foundation (DEB 1655544 and 2137554) to DMA and KS.

Grants from the U.S. National Science Foundation (DEB 1655544 and 2137554).

Additional Editor Comments:

The reviews for your manuscript are now in. While the reviewers appreciate the contributions, they have also raised several major concerns, which must be addressed for further consideration of the manuscript for publication. Please take into full consideration the comments by the reviewers and submit a revised version of your manuscript.

Reviewers' comments:

Reviewer's Responses to Questions

**Comments to the Author**

1. Is the manuscript technically sound, and do the data support the conclusions?

Reviewer #1: Partly

Reviewer #2: Yes

2. Has the statistical analysis been performed appropriately and rigorously? 

Reviewer #1: Yes

Reviewer #2: Yes

3. Have the authors made all data underlying the findings in their manuscript fully available?

Reviewer #1: Yes

Reviewer #2: No

4. Is the manuscript presented in an intelligible fashion and written in standard English?

Reviewer #1: Yes

Reviewer #2: Yes

5. Review Comments to the Author

**Reviewer #1:** This manuscript presents an experimental investigation into the evolutionary responses of genetically modified brewer's yeast to competition and mutualism. The study highlights the impact of biotic interactions, such as competition and mutualism, on growth rate and resource utilization efficiency (RUE) phenotypes. The authors draw the conclusion that the existence of competitive partners decreased RUE and established a significant tradeoff between growth rate and RUE. Conversely, having mutualistic partners resulted in marginally lower growth rates but increased RUE. Furthermore, the authors found that biotic interactions had no significant effect on either growth rate plasticity or RUE plasticity.

This study provides valuable insights into the effects of biotic interactions on trait and plasticity evolution, which is an area that has received limited attention. The Introduction and Discussion sections are well-crafted, effectively summarizing the main points and clearly outlining the study's objectives and findings. However, the Methods and Results sections could benefit from further refinement, particularly with regard to the description of the growth measurements. Moreover, the presented study results and figures are currently insufficient to demonstrate the validity of the reported outcomes convincingly. Therefore, the authors need to revise their manuscript in accordance with the following feedback and recommendations.

## Major comments

1. The growth measurement experiments are poorly described, which poses a significant challenge in comprehending and interpreting the outcomes

a. The Results section does not include a summary of the evolution and growth measurement experiments, making it difficult to interpret the figures without a thorough review of the Methods section.

b. Figure 1A presents some ambiguity regarding the competition between the strains, as only Adenine and Lysine are depicted as being taken up. Additionally, the color codes utilized in the figure lack a clear explanation, further contributing to potential confusion. To ensure a comprehensive understanding of the presented data, it is advisable for the authors to provide a more detailed schematic and a more thorough description.

c. The methodology regarding the growth measurements is somewhat ambiguous. The authors have not specified the number of strains for which growth rates and yields were measured, nor have they provided information regarding which strains were grouped together. Phrases like "When possible, we assayed similar mutualist and competitor strains at the same time" (L248) have left room for interpretation. Given that the description of these experiments is crucial to the study, it would benefit from further refinement.

2. Regarding line 259, it was specified that the yield was computed at the end of the 24-hour interval, as this was when the highest population size was attained throughout the evolutionary experiment. It is critical for the authors to confirm that the low-resource environment is subjected to identical circumstances or, at the very least, ensure that the cells in the culture have not been starved of nutrients for an extended period. Since, the deficiency of adenine and lysine can result in cell death, which may affect OD measurements.

3. According to the authors, the trait and plasticity responses in the LMMs seem to have been minimally affected by strain types. However, it is vital to consider that genetic differences alone could result in varying growth rates among the ancestor strains, which could impact the culture's temporal resource dynamics in various ways (Vidal et al., 2020).

a. The authors did not evaluate how pairing a strain with a different mutant might impact the mutualistic and competitive treatments.

b. Currently, the LMM model does not consider the traits of the partner strain and how it affects the evolution of traits and plasticity. Can the authors provide any insights into this?

4. The source of the data used to produce Figure 3 is unclear, and there is no definitive indication provided in the caption, results, or methods sections as to how the calculations were performed and the values presented were interpreted. Whether these scores represent differences in trait and plasticity values from the growth experiments remains unclear. It is recommended that the authors supplement the results section with a brief explanation to address this issue.

a. The authors may need to reconsider their results as the average effects in most bars are less than the standard errors. What is the source of these errors?

b. Can the authors clarify in the discussion section whether these effects are large enough to affect the outcomes of dynamics in a community?

5. The authors should present a plot with the actual trait and plasticity values for all strains and treatments, either as a main or supplemental figure. This will provide information on the distribution of traits and plasticity values due to different mutations.

6. Figure 4 should be supplemented with a calculation of the Spearman correlation coefficient between the Δgrowth rate and the ΔRUE due to the potential non-linear relationship between the two variables. This addition will provide a more comprehensive understanding of the relationship between Δgrowth rate and ΔRUE and help to identify any strong correlations that may exist.

## Minor comments

7. (Line 351) Fig. 2C does not show growth rate plasticity

8. Line 353's statement, "little difference in growth plasticity," contradicts Line 350's statement, "Growth plasticity was slightly higher."

9. Though Figure 2C shows lower growth rates for partner strains due to mutualistic interactions compared to strains evolved alone, Table 1 (and Figure 3A) indicates no significant difference. Can the authors explain why this is the case?

10. In line 408, the authors refer to Lin et al., 2020 while stating that "a reduction in efficiency is a common response of microbes to serial transfer culturing methods." However, Lin et al., 2020 show that serial dilutions affect growth rates but not yields (RUE). Could the authors clarify this contradiction?

**Reviewer #2:** In this manuscript, Wang et al. have experimentally tested how competition and mutualism affects trait and phenotypic plasticity in pairwise communities of genetically modified brewer’s yeast. They have constructed competitive and mutualistic communities, by combining yeast strains with different mutations in key metabolic genes. They have quantified and compared the evolutionary changes in growth rate, resource use efficiency, and their plasticity in strains evolving alone, with a competitor, and with a mutualist. Their experiments have revealed that strains evolving with a competitive partner had higher growth rates, slightly lower efficiency, and a stronger tradeoff between growth rate and efficiency. In contrast, their results indicate that mutualistic strains had slightly lower growth rates, higher efficiency, and a weak evolutionary tradeoff between growth rate and efficiency. Furthermore, they have observed that competition and mutualism had little effect on plasticity evolution for either trait.

The experiments are well-designed for testing the main hypotheses. The results are significant, and the data supports the main conclusions. However, I have the following minor comments:

1. In lines (172-175), it is mentioned that “Vidal et al (2020) demonstrated that lysine deficient strains die at a faster rate than adenine deficient strains, when starved for the corresponding essential nutrient. This asymmetry in nutrient function may influence evolutionary changes in resource use, growth and population dynamics for different strain types.”.

It is not clear for me how the authors ensure that this asymmetry in nutrient function does not influence their comparisons. Especially, it is mentioned in lines (220-222) that the size of the communities, which are compared in this study is not equal (75 single, 56 mutualistic and 35 competitive communities), because some communities have gone extinct during the 222 experiment. Thus, my main concern here is that the fraction of adenine-deficient-stains vs. lysine-deficient-strains might not stay the same in these communities, and hence the mentioned asymmetry in nutrient function might have acted as a major confounding factor influencing their conclusions. I strongly recommend the authors to clarify and discuss this point.

2. I believe that following sentence in lines (86-88) “Thus, disentangling the importance of biotic interactions from abiotic factors is necessary for understanding how phenotypic plasticity may evolve in general for interacting species” is not well-connected to the preceding sentences, and I think a logical connection is missing here, which might cause confusion. The preceding sentences are focused on competition vs. mutualism, while the final sentence (86-88) suddenly shift the focus towards biotic vs. abiotic interactions, and thus I recommend the authors to rewrite this sentence.

3. It seems one of the abbreviations for the strain names in line 172 is incorrect:

“lysine deficient strains (AdeOP Lys- and Ade- LysWT)”

Ade- LysWT is not a lysine-deficient strain.

6. PLOS authors have the option to publish the peer review history of their article (what does this mean?). If published, this will include your full peer review and any attached files.

Reviewer #1: **Yes: **Dileep Kishore

Reviewer #2: No

---

## [Author Response · Author response to Decision Letter 0]

29 Aug 2024

included in Response to Reviewers' file

---

## [Decision Letter · Decision Letter 1]

15 Sep 2024

PONE-D-24-10370R1Trait and plasticity evolution under competition and mutualism in evolving pairwise yeast communitiesPLOS ONE

Dear Dr. Althoff,

Thank you for submitting your manuscript to PLOS ONE. The manuscript is almost ready for publication, pending some minor suggestions from one of the reviewers, which may be quickly implemented. Please submit your revised manuscript by Oct 30 2024 11:59PM. If you will need more time than this to complete your revisions, please reply to this message or contact the journal office at plosone@plos.org. Please include the following items when submitting your revised manuscript:A rebuttal letter that responds to each point raised by the academic editor and reviewer(s). You should upload this letter as a separate file labeled 'Response to Reviewers'.A marked-up copy of your manuscript that highlights changes made to the original version. You should upload this as a separate file labeled 'Revised Manuscript with Track Changes'.An unmarked version of your revised paper without tracked changes. You should upload this as a separate file labeled 'Manuscript'.If applicable, we recommend that you deposit your laboratory protocols in protocols.io to enhance the reproducibility of your results. Protocols.io assigns your protocol its own identifier (DOI) so that it can be cited independently in the future. For instructions see: https://journals.plos.org/plosone/s/submission-guidelines#loc-laboratory-protocols. Additionally, PLOS ONE offers an option for publishing peer-reviewed Lab Protocol articles, which describe protocols hosted on protocols.io. Read more information on sharing protocols at https://plos.org/protocols?utm_medium=editorial-email&utm_source=authorletters&utm_campaign=protocols.

We look forward to receiving your revised manuscript.

Kind regards,

Karthik Raman, Ph.D.

Academic Editor

PLOS ONE

Journal Requirements:

Reviewers' comments:

Reviewer's Responses to Questions

**Comments to the Author**

1. If the authors have adequately addressed your comments raised in a previous round of review and you feel that this manuscript is now acceptable for publication, you may indicate that here to bypass the “Comments to the Author” section, enter your conflict of interest statement in the “Confidential to Editor” section, and submit your "Accept" recommendation.

Reviewer #1: All comments have been addressed

Reviewer #2: All comments have been addressed

2. Is the manuscript technically sound, and do the data support the conclusions?

Reviewer #1: Yes

Reviewer #2: Yes

3. Has the statistical analysis been performed appropriately and rigorously? 

Reviewer #1: Yes

Reviewer #2: Yes

4. Have the authors made all data underlying the findings in their manuscript fully available?

Reviewer #1: Yes

Reviewer #2: (No Response)

5. Is the manuscript presented in an intelligible fashion and written in standard English?

Reviewer #1: Yes

Reviewer #2: Yes

6. Review Comments to the Author

Reviewer #1: The manuscript presents a valuable study illustrating how biotic interactions, including both competition and mutualism, influence the growth rate and resource utilization efficiency (RUE) phenotypes within a microbial community. The authors have modified the manuscript in response to the feedback and suggestions provided by the reviewers, effectively addressing most of the reviewers’ concerns. The revisions incorporated into the Results section, along with the added supplementary material, have notably enhanced the clarity and comprehensiveness of the study. Below are a few final comments:

## Major comments

In Line 225 the authors mention that the “Extinction of paired strain communities was due to a combination of differences in growth rates among Ade- and Lys- strains, the serial dilution process itself, and interaction type”. However, based on Figure 3, the growth rate and RUE of the CA2 strain are similar to those of CA1, and likewise, the values for CL1 are comparable to CL2. In light of this, it would be beneficial for the authors to offer an explanation regarding the potential factors that led to the extinction of the CA2 strain.

## Minor comments

1) Figure 1: the authors could consider including an annotation for Figure 1a, analogous to the annotation present in Figure 1b, such as: “media contains Ade and Lys, and the strains compete for all other nutrients”

2) Figure 3: The quality of the image is poor and appears blurry

3) (Line 343) “growth rate at 4, 6 and 24 hours”. It might be more suitable to use 'yield' instead of 'growth rate'.

4) (Line 347-348) Is this referring to Figures 2a and 2b instead of Figures 3a and 3b? Figure 3 does not compare low and high resource environments.

5) (Line 359-361) Figure 2 does not show growth rate plasticity or RUE plasticity

6) (Line 368) Change Figure 4a to Figure 4, since 4a only displays the growth rate

Reviewer #2: (No Response)

7. PLOS authors have the option to publish the peer review history of their article (what does this mean?). If published, this will include your full peer review and any attached files.

Reviewer #1: **Yes: **Dileep Kishore

Reviewer #2: **Yes: **Sayed-Rzgar Hosseini

---

## [Author Response · Author response to Decision Letter 1]

20 Sep 2024

see Response to Reviewers_R2 file

---

## [Editor Report · Decision Letter 2]

24 Sep 2024

Trait and plasticity evolution under competition and mutualism in evolving pairwise yeast communities

PONE-D-24-10370R2

Dear Dr. Althoff,

We’re pleased to inform you that your manuscript has been judged scientifically suitable for publication and will be formally accepted for publication once it meets all outstanding technical requirements.

Kind regards,

Karthik Raman, Ph.D.

Academic Editor

PLOS ONE
---

## [Editor Report · Acceptance letter]

1 Oct 2024

PONE-D-24-10370R2 

PLOS ONE

Dear Dr. Althoff, 

I'm pleased to inform you that your manuscript has been deemed suitable for publication in PLOS ONE. Congratulations! Your manuscript is now being handed over to our production team.

Kind regards, 

on behalf of

Dr. Karthik Raman 

Academic Editor

PLOS ONE